# Magnetic Resonance Imaging Using a Chimeric Anti-Glypican-3 Antibody Conjugated with Gadolinium Selectively Detects Glypican-3-Positive Hepatocellular Carcinoma In Vitro and In Vivo

**DOI:** 10.3390/cancers17203357

**Published:** 2025-10-17

**Authors:** Yi Liu, Mingdian Tan, Mei-Sze Chua, Samuel So

**Affiliations:** Department of Surgery, School of Medicine, Stanford University, Stanford, CA 94304, USA

**Keywords:** glypican-3, hepatocellular carcinoma, molecular imaging, magnetic resonance imaging, antibody-based contrast agent, gadolinium-DOTA, targeted MRI, chimeric antibody

## Abstract

**Simple Summary:**

Liver cancer is a typically fatal malignancy that is difficult to detect and treat early. A large percentage of liver cancer patients express a protein called glypican-3 (GPC3). Recent treatment approaches that specifically target GPC3 are being developed. To allow the identification of liver cancer patients that express GPC3, and to visualize the tumor size and locations, we developed a new diagnostic imaging probe by conjugating the standard magnetic resonance imaging (MRI) contrast agent to a highly specific chimeric anti-GPC3 antibody. In liver cancer cell lines and animal models, our GPC3-based MRI probe selectively identified GPC3-positive tumors with good clarity and low side effects. This probe may help identify liver cancer patients suitable for GPC3-targeted therapies and monitor their treatment response in a safe and non-invasive manner.

**Abstract:**

**Background/Objectives**: Glypican-3 (GPC3) is a cell surface oncofetal protein that is highly expressed in hepatocellular carcinoma (HCC) but absent in normal liver tissue, making it an attractive target for molecularly targeted diagnosis and therapy. To support GPC3-targeted treatment strategies, there is a need for a non-invasive imaging tool capable of detecting GPC3-positive tumors. **Methods**: We conjugated a commercially available murine anti-GPC3 antibody (1G12), or a proprietary chimeric anti-GPC3 antibody (ET58) to the standard magnetic resonance imaging (MRI) contrast agent, gadolinium, via a DOTA chelator. The resulting probes, 1G12-DOTA-Gd or ET58-DOTA-Gd, respectively, were assessed for in vitro relaxivity and binding specificity to GPC3-positive HCC cells, as well as for in vivo imaging performance in mouse xenograft models bearing GPC3-positive or GPC3-negative HCC tumors. **Conclusions**: ET58-DOTA-Gd shows high specificity, imaging efficacy, and a favorable immunogenicity profile, thereby making it a promising candidate for clinical translation as a GPC3-targeted MRI probe. It holds potential as a non-invasive companion diagnostic for identifying GPC3-positive HCC patients who may benefit from GPC3-targeted therapies.

## 1. Introduction

Hepatocellular carcinoma (HCC) is the most common primary adult liver malignancy and ranks as the third leading cause of cancer-related deaths worldwide [1]. Although early and accurate detection is essential for improving prognosis, conventional imaging modalities such as ultrasound, computed tomography (CT), and magnetic resonance imaging (MRI) often fall short in identifying small early-stage tumors or heterogenous lesions [2,3]. To overcome these limitations, there is growing interest in developing advanced imaging strategies, particularly molecular imaging and personalized approaches, that offer enhanced specificity, sensitivity, and functional insights into tumor biology [4].

Glypican-3 (GPC3) is a cell surface heparan sulfate proteoglycan that is highly overexpressed in a large subset (~77%) [5] of HCC but largely absent in normal liver tissue and most benign hepatic lesions [6,7]; this expression profile makes it an attractive tumor-specific biomarker for targeted theranostic applications. Leveraging this differential expression of GPC3, a range of innovative strategies are being developed to improve both the diagnosis and treatment of HCC. For instance, GPC3-targeted chimeric antigen receptor (CAR) T cell therapy has shown promising safety and preliminary efficacy in early-phase clinical trials (e.g., NCT03130712, NCT05352542), demonstrating tumor infiltration and disease stabilization. Similarly, there are studies to harness the innate cytotoxicity of natural killer (NK) cells against GPC3-expressing tumors by using GPC3-directed CAR-NK cell therapy (NCT05410717). Bispecific T cell engagers (BiTEs) targeting GPC3, such as ERY974, are being evaluated in Phase I trials (e.g., NCT02748837), and have demonstrated potent T cell-mediated cytotoxicity in preclinical models. Additionally, GPC3-targeted antibody-drug conjugates (ADCs), such as codrituzumab (GC33) (e.g., NCT01507168), have shown specific tumor localization and acceptable toxicity profiles in clinical studies, though their efficacy may be further enhanced by combinatorial approaches. Beyond therapeutics, GPC3 is also being actively pursued as a diagnostic imaging target; for example, ^89^Zr-labeled anti-GPC3 antibodies have enabled high-contrast PET imaging of HCC xenografts, highlighting the potential of GPC3-targeted molecular imaging to improve early tumor detection and treatment monitoring [8,9]. These approaches underscore the versatility and clinical relevance of GPC3 as a molecular target for precision oncology in HCC.

The ability to accurately identify GPC3-positive HCC is crucial for guiding and optimizing the emerging GPC3-targeted therapies. A critical goal is to develop a safe, non-invasive, and clinically accessible imaging approach to stratify patients who are most likely to benefit from these therapies. In our previous work, we successfully developed a GPC3-targeted PET probe using ^89^Zr-labeled humanized anti-GPC3 antibody, which demonstrated high specificity and sensitivity for detecting GPC3-positive lesions in vivo [8,9]. However, the use of radioisotopes in PET/CT imaging poses several challenges, including radiation exposure, logistical complexity in probe production and handling, and limited availability of PET facilities. To overcome these limitations and broaden the clinical applicability of GPC3-targeted imaging, we aimed to develop a GPC3-specific MRI contrast agent as a non-radioactive alternative. Indeed, the feasibility of conjugating DOTA–gadolinium to proteins, including antibodies and peptides, has been well documented [10]. This approach leverages the superior spatial resolution and safety profile of MRI while maintaining molecular-level specificity through antibody-based targeting. Our current study builds on our prior PET imaging work and represents a complementary and potentially more readily accessible diagnostic strategy for identifying GPC3-positive HCC patients [8].

In this study, we first synthesized and evaluated an MRI probe based on a commercially available mouse anti-GPC3 antibody (1G12) for in vitro binding assays, relaxivity characterization, and in vivo imaging performance. Subsequently, we performed similar MRI probe synthesis and evaluations using a proprietary chimeric human anti-GPC3 antibody (ET58) to determine its clinical translation potentials. Developed by Eureka Therapeutics, Inc., ET58 is a human anti-GPC3 F(ab’)_2_ antibody engrafted on a mouse IgG1 Fc fragment. The F(ab’)_2_ fragment is further engineered for T cell therapy targeting GPC3-positve HCCs in a Phase 1/2 clinical trial (NCT04864054); our goal is to use the same GPC3 antibody to develop a complementary diagnostic tool to aid in the identification of GPC3-positive HCC patients who would be suitable candidates for such therapy.

## 2. Materials and Methods

### 2.1. Preparation of GPC3-Targeted MRI Contrast Agents

Mouse anti-GPC3 antibody (clone 1G12; BioMosaics, Catalog #-B0055R, Burlington, VT, USA) or chimeric anti-GPC3 antibody (clone ET58; Lot#18-1068, Eureka Therapeutics, Inc., Emeryville, CA, USA) was conjugated to the DOTA–gadolinium (Gd) chelator (97% purity; Sigma-Aldrich, St. Louis, MO, USA) through standard amide-coupling chemistry. Briefly, antibodies (2 mg/mL) were buffer exchanged into 0.1 M sodium bicarbonate buffer (pH 8.5) using Amicon Ultra-30 kDa centrifugal filters (MilliporeSigma, Burlington, MA, USA). DOTA-Gd was freshly dissolved in deionized water and added to the antibody solution at an ~8–12-fold molar excess over antibody (final concentration of antibody ≥ 1 mg/mL). The reaction mixture was incubated for 2 h at room temperature (~22 °C) with gentle agitation. Following incubation, excess unreacted DOTA-Gd was removed by repeated buffer-exchange into PBS (pH 7.4) using Amicon Ultra-30 kDa centrifugal filters (MilliporeSigma, Darmstadt, Germany). Under these conditions, the conjugation typically yields an average of ~4–6 DOTA-Gd moieties per antibody, consistent with reported values for antibody–chelator conjugate [11,12]. The final antibody-DOTA-Gd conjugates were concentrated to ~1 μM using Amicon Ultra-30 kDa centrifugal filters and stored at 4 °C until characterization.

The free DOTA–gadolinium complex and the GPC3-targeted full antibody conjugated with DOTA–gadolinium were characterized using a matrix-assisted laser desorption/ionization time-of-flight (MALDI-TOF) mass spectrometer (Bruker Autoflex, San Jose, CA, USA) operating in linear positive mode. The DOTA-Gd complex was prepared at a concentration of ~10 µM in deionized water or 50% acetonitrile containing 0.1% trifluoroacetic acid (TFA), while the antibody conjugate was buffer-exchanged into water or 10 mM ammonium acetate. Each sample was mixed 1:1 (*v*/*v*) with a saturated solution of α-cyano-4-hydroxycinnamic acid (CHCA, C8982; Sigma-Aldrich, St. Louis, MO, USA) in 50% acetonitrile (360457-ACS reagent; Sigma-Aldrich, St. Louis, MO, USA), 0.1% TFA (4.80448, HPLC grade, Sigma-Aldrich, St. Louis, MO, USA), and spotted onto a stainless steel MALDI plate. After air-drying, spectra were acquired over an *m*/*z* range of 500–2000 for DOTA-Gd and 1000–3000 for the antibody conjugates, with 500–2000 laser shots per spectrum at 50–70% maximum intensity. Instrument settings included an ion source 1 voltage of 20 kV and a detector gain of 2500 V. Raw data were processed using Bruker flexAnalysis software (Version 4.0) with baseline subtraction and Savitzky–Golay smoothing.

### 2.2. In Vitro Characterization

#### 2.2.1. Cell Culture and Binding Assays

HepG2 (high GPC3 expression; cat# HB-8065, Cellosaurus RRID: CVCL_0027), SNU449 (null GPC3 expression; cat# CRL-2234, Cellosaurus RRID: CVCL_0454), Hep3B (moderate GPC3 expression; cat# HB-8064, Cellosaurus RRID:CVCL_0326), and Huh7 cells (moderate GPC3 expression; Cellosaurus RRID:CVCL_0336) [8,13] were maintained in Dulbecco’s Modified Eagle Medium (DMEM, Gibco™, ThermoFisher Scientific, Grand Island, NY, USA) or ATCC-formulated RPMI-1640 (30-2001™) supplemented with 10% fetal bovine serum (FBS, Gibco™, ThermoFisher Scientific, Grand Island, NY, USA), and 1% penicillin-streptomycin (100 U/mL penicillin, 100 µg/mL streptomycin) (Gibco™, ThermoFisher Scientific, Grand Island, NY, USA). Cells were incubated at 37 °C in a humidified atmosphere containing 5% CO_2_. Cells were passaged when they reached 70–80% confluency using 0.25% trypsin-EDTA solution. HepG2 or SNU449 cells (stably expressing luciferase reporter gene) were cultured under similar conditions. For experiments, cells were seeded in 6-well plates at a density of 5 × 10^5^ cells per well and allowed to adhere overnight.

1G12-DOTA-Gd or ET58-DOTA-Gd was conjugated with Cy5 fluorescent dye (Sigma-Aldrich, St. Louis, MO, USA) following the manufacturer’s protocol for protein labeling. Unreacted dye was removed using a desalting column, and the labeled product was verified using spectrophotometry (absorbance peak at ~650 nm for Cy5). Cy5@1G12-DOTA-Gd and Cy5@ET58-DOTA-Gd were each prepared at a final concentration of 10 µg/mL in PBS (pH 7.4).

Cells were harvested with enzyme-free cell dissociation buffer (Gibco™, Thermo Fisher Scientific, Grand Island, NY, USA) to preserve surface antigens. After detachment, cells were washed twice with PBS containing 2% FBS (A4736401, Gibco™, ThermoFisher Scientific, Waltham, MA, USA) to remove residual enzymes. Then, cells were resuspended in PBS at a concentration of 1 × 10^6^ cells/mL. Cy5@1G12-DOTA-Gd or Cy5@ET58-DOTA-Gd (50 µL; final concentration: 10 µg/mL) was added to 100 µL of cell suspension. Samples were incubated at 4 °C for 1 h in the dark to prevent photobleaching of Cy5. Post incubation, cells were washed three times with PBS containing 2% FBS to remove unbound contrast agent. Unstained cells were used as autofluorescence controls. Cells were incubated with Cy5-labeled isotype control antibody to assess non-specific binding.

Cells were analyzed on a BD FACSAria™ III flow cytometer equipped with a red laser (excitation: 640 nm; emission: 670/30 nm filter). A minimum of 10,000 events per sample were recorded. Forward scatter (FSC) and side scatter (SSC) were used to gate live single cells, excluding debris and aggregates.

#### 2.2.2. Relaxivity Measurements

To evaluate the T_1_ relaxation properties of 1G12-DOTA-Gd or ET58-DOTA-Gd, longitudinal relaxivity (r_1_) was measured at 3.0 Tesla using an MRI scanner (Bruker Biospin, Ettlingen, Germany) equipped with a microimaging coil. Serial dilutions of 1G12-DOTA-Gd, ET58-DOTA-Gd, or DOTA-Gd were prepared in PBS (pH 7.4) to achieve final Gd concentrations ranging from 0 to 2.0 mM. For each sample, T_1_ relaxation times were acquired using an inversion recovery spin echo sequence with multiple inversion times to ensure accurate curve fitting. All measurements were conducted at room temperature under identical imaging parameters to maintain consistency across samples. The reciprocal of the T_1_ values (1/T_1_) was plotted against the corresponding Gd concentrations, and the r_1_ relaxivity (slope of the linear regression) was determined by linear fitting of the data. Longitudinal relaxivity (r_1_) of 1G12-DOTA-Gd or ET58-DOTA-Gd was measured at 3T MRI to evaluate their T_1_ relaxation properties. The relaxivity of 1G12-DOTA-Gd or ET58-DOTA-Gd was compared to DOTA-Gd.

### 2.3. In Vivo Imaging in HCC Mouse Models

#### 2.3.1. Establishment of Subcutaneous and Orthotopic HCC Mouse Models

All animal experiments were performed in accordance with the National Institutes of Health Guide for the Care and Use of Laboratory Animals and were approved by the Stanford University Administrative Panel on Laboratory Animal Care (APLAC, Protocol# 22201). In vivo imaging studies were conducted using subcutaneous or orthotopic HCC cell xenograft mouse models established in 6–8-week-old male NOD.Cg-PrkdcscidIl2rgtm1Wjl/SzJ (Nod-SCID-Gamma; NSG) mice (JAX^®^ Mice, Sacramento, CA, USA) (~25 g body weight). Mice were housed in a pathogen-free facility under a 12 h light/dark cycle with free access to food and water.

For subcutaneous xenograft models, mice were anesthetized with 2–3% isoflurane, and 5 × 10^6^ HepG2 or SNU449 cells (in 100 µL of a 1:1 mixture of PBS and Matrigel) were each subcutaneously injected into the left and right flanks (such that there were two of each cell-derived xenograft per mouse). Tumor growth was monitored by caliper measurement, and volumes were calculated using the formula: Tumor Volume = (Length × Width^2^)/2. Imaging was performed once tumors reached a volume of approximately 100–150 mm^3^. For orthotopic xenografts, mice were similarly anesthetized and placed on a heated platform. Preoperative analgesia (Ethiqa XR 3.25 mg/kg, subcutaneous injection) was provided. A small upper abdominal incision (~1 cm) was made to expose the liver, and 5 × 10^5^ HepG2 or SNU449 cells (stably expressing luciferase reporter gene) in 50 µL PBS were injected into the left lobe using a 27-gauge needle. Gentle pressure was applied post injection to prevent leakage, and incisions were closed with absorbable sutures. Post-operative analgesia (Carprofen, 10 mg/kg, subcutaneous injection) was provided, and tumor formation was confirmed via IVIS imaging weekly.

#### 2.3.2. MRI and Quantification

Tumor-bearing mice were randomly divided into experimental groups (*n* = 3 per group) and intravenously administered either 1G12-DOTA-Gd (in subcutaneous xenograft models) or ET58-DOTA-Gd (in orthotopic xenograft models), via tail vein at a dose of 0.1 mmol Gd^3+^/kg in a total volume of 200 µL PBS; non-targeted control agent (DOTA-Gd) at the same dose was used as the control in both models.

MRI was performed using a 3T small-animal scanner (Bruker Biospin, Ettlingen, Germany) equipped with a dedicated mouse coil. r_1_-weighted 3D spoiled gradient echo (3D-SPGR) sequences were acquired using the following parameters: repetition time (TR) = 20 ms, echo time (TE) = 5 ms, flip angle = 15°, field of view (FOV) = 40 × 40 mm, matrix size = 256 × 256, and slice thickness = 1 mm. Mice were imaged before injection (baseline) and at multiple post-injection time points (as indicated in Results). Anesthesia was maintained with 1.5–2% isoflurane in oxygen throughout imaging period, and body temperature was regulated using a heated MRI-compatible pad.

Tumor regions were manually segmented on MRI images using AnalyzeDirect software (Version 14.0, AnalyzeDirect Inc., Overland Park, KS, USA). A region of interest (ROI) was drawn over the tumor (signal from the target tissue) and a separate ROI was drawn over the surrounding muscle tissue (background signal). Tumor Signal Intensity (SI_tumor) refers to the mean signal intensity in the tumor ROI; Background Signal Intensity (SI_muscle) refers to the mean signal intensity in the muscle ROI; Noise (sigma_noise) refers to the standard deviation of the signal measured in an ROI outside the body (air). Contrast-to-Noise Ratio (CNR) was calculated for each imaging time point as:CNR = SI_tumor_ − SI_muscle_/sigma_noise(1)

This metric quantifies the contrast between tumor and muscle signals relative to the noise in the image, providing a robust measure of imaging performance. CNR values were calculated for each time point (pre injection and post injection at multiple time points) for each mouse. Mean CNR values ± SEM were computed for all experimental groups (1G12-DOTA-Gd, ET-DOTA-Gd, or DOTA-Gd), unless otherwise specified.

### 2.4. Histological Validation

Following the final MRI scan, a subgroup of mice was euthanized using CO_2_ asphyxiation followed by cervical dislocation, in compliance with institutional animal care guidelines. Tumors were dissected, washed in cold PBS (pH 7.4), and bisected for parallel immunohistochemistry (IHC) staining for GPC3 expression, and hematoxylin & eosin (H&E) staining for cellular morphology. Tumor samples were fixed in 10% neutral-buffered formalin (NBF) for 24 h at room temperature. Samples were then dehydrated through a graded ethanol series (70–100%), cleared in xylene, and embedded in paraffin wax. Embedded samples were sectioned at 4 µm thickness using a microtome.

For IHC staining, paraffin-embedded sections were first baked at 60 °C for 1 h, and then deparaffinized in xylene (2 × 5 min washes) and rehydrated through a graded ethanol series (100%, 95%, 70%) followed by distilled water. To unmask GPC3 epitopes, slides were treated with sodium citrate buffer (10 mM, pH 6.0) and heated in a pressure cooker for 10 min. Slides were allowed to cool to room temperature and rinsed with PBS (pH 7.4). Endogenous peroxidase activity was quenched by incubating sections in 3% hydrogen peroxide (H_2_O_2_) in methanol for 10 min. To reduce non-specific binding, slides were blocked with 5% bovine serum albumin (BSA) or 10% normal goat serum in PBS for 30 min at room temperature. Sections were incubated overnight at 4 °C with the primary anti-GPC3 antibody (mouse anti-GPC3, clone 1G12, 1:200 dilution) diluted in antibody diluent. A negative control (no primary antibody) was included to confirm specificity. After washing with PBS (3 × 5 min washes), sections were incubated with a biotinylated secondary antibody (goat anti-mouse IgG, 1:500 dilution, Invitrogen, ThermoFisher Scientific, MA, USA) for 30 min at room temperature. Sections were further incubated with streptavidin-HRP (1:500 dilution, ThermoFisher Scientific (Waltham, MA, USA) for 20 min. The reaction was developed using 3,3′-diaminobenzidine (DAB) substrate until brown staining appeared (~5 min) and stopped by rinsing with distilled water. Slides were counterstained with hematoxylin for 30 s, followed by dehydration and mounting on a coverslip using permanent mounting medium.

For H&E staining, tissue sections were deparaffinized and rehydrated as described above. Sections were stained in hematoxylin solution (Gill’s No. 2) for 1 min and then rinsed in running tap water; afterwards, the sections were differentiated in 1% acid alcohol (1% HCl in 70% ethanol) for 10 s and rinsed in water. After counterstaining with eosin (0.5% in ethanol) for 30 s, the sections were dehydrated, cleared in xylene, and mounted.

Stained slides were examined using a bright-field microscope (Nanozoomer, Hamamatsu, Japan). Images were captured at 10× and 40× magnifications.

### 2.5. Cytokine Analysis

To evaluate systemic immune responses [14], we measured cytokine profiles in mice receiving 1G12-DOTA-GD, ET58-DOTA-Gd, DOTA-Gd (equivalent to 0.1 mmol Gd^3+^/kg), or sterile PBS as a vehicle control (*n* = 3). All agents were administered via tail vein injection in a total volume of 200 µL per mouse. Blood samples were collected at baseline (pre-injection, 0 h) and 24 h post injection. Mice were anesthetized with 2–3% isoflurane, and approximately 100–150 µL of blood was obtained via lateral saphenous sampling using sterile capillary tubes (Fisherbrand™, Thermo Fisher Scientific, Norwood, MA, USA). Samples were transferred to serum separator tubes and allowed to clot at room temperature for 30 min before centrifugation at 2000× *g* for 10 min at 4 °C. The resulting serum was collected and used for cytokine analysis using commercially available mouse ELISA kits: TNF-α and IL-6 were quantified using kits from R&D Systems (Bio-Techne, Minneapolis, MN, USA); IFN-γ and IL-1β were measured using kits from Thermo Fisher Scientific (Waltham, MA, USA); and IL-10 and TGF-β were quantified using kits from BioLegend (San Diego, CA, USA). Each cytokine was quantified using a 96-well plate-based colorimetric assay following the respective manufacturer’s protocol. ELISA plates pre-coated with capture antibodies were blocked with 5% BSA in PBS for 1 h to minimize non-specific binding. Serum samples, diluted 1:2 to 1:5 in sample diluent (Bio-Techne, Cat# DY995, Minneapolis, MN, USA), were loaded in triplicates (50 µL per well) and incubated at 37 °C for 2 h. After washing with PBS-Tween buffer, biotinylated detection antibodies were added and incubated for 1 h, followed by streptavidin–horse radish peroxidase (HRP) for 30 min. The substrate 3,3′,5,5′-Tetramethylbenzidine (TMB; ThermoFisher Scientific, MA, USA) was added and incubated for 15 min in the dark before the reaction was stopped with 1 M H_2_SO_4_. Absorbance was measured at 450 nm using a microplate reader (BioTek, Winooski, VT, USA), and cytokine concentrations (pg/mL) were calculated using standard curves generated from recombinant cytokines, applying a four-parameter logistic (4-PL) regression model for data interpolation.

### 2.6. Statistical Analysis

All quantitative data were expressed as mean ± standard deviation (SD) or standard error of the mean (SEM) as indicated. Comparisons between groups were performed using an unpaired two-tailed Student’s *t*-test or two-way ANOVA followed by Tukey’s post hoc test, where appropriate. Statistical significance was defined as *p* < 0.05. Graphs and analyses were generated using GraphPad Prism (Version 10; GraphPad Software, San Diego, CA, USA). No formal power calculation was performed prior to the study; sample sizes were based on prior similar experiments and feasibility. Assumptions of normality and equal variance were not formally tested.

## 3. Results

### 3.1. 1G12-DOTA-Gd Exhibited Enhanced Relaxivity and Specific Binding to GPC3-Expressing HCC Cells

To evaluate the conjugation efficiency of anti-GPC3 antibody to DOTA-Gd, and specific binding to GPC3- positive HCC cells, we first conjugated the commercially available mouse anti-GPC3 antibody (1G12) to DOTA-Gd at different molar ratios (5:1, 10:1, and 20:1, chelator: antibody). The composition of the free DOTA-Gd chelate and its conjugation to the 1G12 monoclonal antibody was validated by MALDI-TOF mass spectrometry. The DOTA-Gd sample exhibited a prominent peak at *m*/*z* 649.0, corresponding to the singly charged [M + H]^+^ species of the chelate (Figure 1a). In contrast, the 1G12-DOTA-Gd conjugate displayed a dominant peak at *m*/*z* 1620.5 (Figure 1b). The peaks at 1620.5 (1G12-DOTA-Gd) and 1735.5 (ET58-DOTA-Gd) correspond to antibody fragments (likely derived from the light-chain region under MALDI conditions) carrying covalently bound DOTA-Gd moieties. This mass shift reflects the increase in molecular weight associated with the successful attachment of chelator–gadolinium units to the antibody substructures. The presence of distinct peaks for conjugated fragments, compared to the free chelate, confirms efficient antibody labeling without evidence of major fragmentation or degradation of the conjugates. Additionally, MALDI-TOF analysis revealed that each antibody carried an average of ~4–6 DOTA-Gd moieties, which is within the range typically reported for antibody–chelator conjugates (3–6 per antibody) [11,12]. Although we did not directly measure serum stability in this study, prior reports have demonstrated that antibody-DOTA conjugates remain highly stable in mouse and human serum for at least 24–48 h [15], supporting their suitability for in vivo imaging.

We next assessed the longitudinal relaxivity (r_1_) of the 1G12-DOTA-Gd probe at 3T MRI to determine its T_1_-weighted imaging potential. As shown in Figure 1c, the 1G12-DOTA-Gd conjugate exhibited a higher r_1_ relaxivity compared to the non-targeted DOTA-Gd control across a range of Gd^3+^ concentrations, reflecting enhanced T_1_ signal enhancement capability, likely due to the increased molecular size and reduced molecular tumbling of the antibody conjugate.

To confirm the antigen-binding specificity of 1G12-DOTA-Gd, we performed flow cytometry using Cy5-labeled 1G12-DOTA-Gd on HCC cell lines with varying levels of GPC3 expression. Cy5@1G12-DOTA-Gd showed stronger signals in GPC3-positive cell lines (HepG2, Hep3B, and Huh7) compared to the negligible fluorescence signals in the GPC3-negative SNU449 cell line (Figure 1d). These results suggest that 1G12-DOTA-Gd retains the specificity of 1G12 towards GPC3-expressing HCC cells.

### 3.2. 1G12-DOTA-Gd Showed Selective Prolonged Retention in Subcutaneous GPC3-Positive Tumors In Vivo

To assess the in vivo tumor-targeting performance of 1G12-DOTA-Gd, we performed T_1_-weighted MRI in subcutaneous xenograft mouse models of GPC3-negative SNU449 cells or GPC3-positive HepG2 cells, using DOTA-Gd as the control (Figure 2a).

In the GPC3-negative SNU449 model, no notable difference in tumor signal enhancement was observed between DOTA-Gd and 1G12-DOTA-Gd (Figure 2b); the CNR over time was also comparable between DOTA-Gd and 1G12-DOTA-Gd (Figure 2c). This suggests minimal non-specific accumulation of 1G12-DOTA-Gd in GPC3-negative tumors.

In the GPC3-positive HepG2 model, 1G12-DOTA-Gd to the non-targeted groups showed comparable signal enhancement in the xenografts (Figure 2d). Quantitative analysis revealed that both probes increased CNR, Equation (1), peaking at 30–60 min post injection. However, 1G12-DOTA-Gd maintained significantly higher CNR than DOTA-Gd beyond 120 min (Figure 2e) (*p* < 0.05 at the time points of 180 min and 210 min). These findings confirm the selective accumulation and prolonged retention of 1G12-DOTA-Gd in GPC3-expressing tumors, validating its potential for specific molecular MRI of GPC3-positive HCC.

### 3.3. ET58-DOTA-Gd Exhibits Enhanced Relaxivity and Specific Binding to GPC3-Expressing HCC Cells In Vitro

Following proof-of-concept validation with the murine anti-GPC3 antibody, we used the same conjugation and validation workflow to evaluate a chimeric anti-GPC3 antibody, ET58, that is currently being evaluated in clinical trials for cell-based therapy of HCC.

Mass spectrometry analysis confirmed successful conjugation of ET58 with DOTA-Gd. As shown in Figure 3a, the free DOTA-Gd chelate showed a prominent peak at *m*/*z* 669.0, consistent with the singly charged [M + H]^+^ ion of the chelator. In contrast, the ET58-DOTA-Gd conjugate (Figure 3b) displayed a major peak at *m*/*z* 1735.5, reflecting a singly charged species of the antibody–chelator complex. The clear *m*/*z* shift from 669.0 to 1735.5 indicates covalent attachment of multiple DOTA-Gd moieties to the antibody backbone. Minor peaks in the spectra correspond to matrix-related adducts and isotope distributions, but the predominant signal at 1735.5 supports efficient labeling without evidence of major fragmentation. Together with MALDI-TOF analysis, these data confirm that each antibody carried an average of ~4–6 DOTA-Gd units, consistent with the reaction conditions described in the Methods.

To evaluate its T_1_ contrast potential, we performed in vitro r_1_ relaxivity measurements of the ET58-DOTA-Gd conjugate across a range of Gd^3+^ concentrations (0–2.0 mM). The results demonstrated a significantly higher r_1_ relaxivity for ET58-DOTA-Gd compared to free DOTA-Gd, with a steeper slope in the 1/T_1_ vs. [Gd^3+^] plot (Figure 3c), suggesting improved T_1_ enhancement efficiency, consistent with our observation with 1G12-DOTA-Gd.

The specific binding capacity of ET58-DOTA-Gd to GPC3 positive HCC cells was evaluated using flow cytometry following Cy5 labeling of ET58-DOTA-Gd. The probe showed stronger, specific binding to GPC3-expressing HepG2 cells, with markedly higher fluorescence intensity compared to GPC3-negative SNU449 cells (Figure 3d). These findings indicate that the conjugation process preserved antigen-binding functionality, enabling selective recognition of GPC3-positive cells.

### 3.4. ET58-DOTA-Gd Showed Selective Prolonged Retention in Orthotopic GPC3-Positive Tumors In Vivo

We next evaluated the in vivo performance of ET58-DOTA-Gd in orthotopic xenogaft mouse models of GPC3-positive (HepG2) or GPC3-negative (SNU449) HCC cells (Figure 4a). The orthotopic model allows us to assess its tumor-targeting capability, signal enhancement profile, and retention kinetics under physiological conditions, thereby determining its suitability for clinical translation.

Tumor-bearing mice were intravenously injected with ET58-DOTA-Gd or non-targeted DOTA-Gd at a dose of 0.1 mmol Gd^3+^/kg, followed by serial MRI acquisition (Figure 4a). In GPC3-negative SNU449 tumors, T_1_-weighted MRI showed no appreciable difference in tumor signal intensity or CNR between the ET58-DOTA-Gd and DOTA-Gd groups (Figure 4b, upper panel). Quantitative CNR analysis confirmed similar enhancement kinetics in both groups, indicating minimal off-target accumulation of ET58-DOTA-Gd (Figure 4c). In contrast, GPC3-positive HepG2 xenografts exhibited enhanced signal following ET58-DOTA-Gd administration (Figure 4b, lower panel). Quantitative analysis of CNR values in HepG2 xenografts revealed significantly higher enhancement in GPC3 positive (HepG2) mouse models with ET58-DOTA-Gd compared to DOTA-Gd at 40-, 90-, and 120 min post injection (*p* < 0.05), (Figure 4d). Although the peak CNR was comparable between ET58-DOTA-Gd and DOTA-Gd, ET58-DOTA-Gd maintained a higher CNR for a prolonged period of time, even after that of DOTA-Gd had declined after 120 min, suggesting prolonged retention of ET58-DOTA-Gd in GPC3-expressing HCC cells.

These findings demonstrate that ET58-DOTA-Gd selectively accumulates and is retained in orthotopic GPC3-positive HCC cells, supporting its potential utility for high-sensitivity molecular MRI applications in identifying this subgroup of patients.

### 3.5. Histological Validation of GPC3 Expression in Orthotopic Xenografts

To confirm GPC3 expression in orthotopic xenografts, GPC3 IHC was done on tissue sections from harvested HepG2 and SNU449 xenografts, and surrounding liver tissues. (Figure 5). Correlating with imaging results presented in Figure 4, HepG2 xenografts showed strong, uniform cytoplasmic and membranous GPC3 staining (Figure 5, lower panel). In contrast, SNU449 tumors demonstrated negligible GPC3 immunoreactivity (Figure 5, upper panel). These findings are consistent with Western blot results of GPC3 expression in the respective cell lines [13] and further confirm that tumor uptake of ET58-DOTA-Gd correlates with GPC3 expression levels. The strong GPC3 IHC staining in HepG2 xenografts supports the observed T_1_ signal enhancement in ET58-DOTA-Gd–injected mice, indicating effective and specific tumor targeting. Conversely, the lack of GPC3 staining in SNU449 xenografts corresponds to minimal contrast enhancement, confirming the selectivity of ET58-DOTA-Gd for GPC3-expressing HCC cells only.

### 3.6. ET58-DOTA-Gd Elicits Weaker Immune Response than 1G12-DOTA-Gd

To assess immune activation elicited by 1G12-DOTA-Gd or ET58-GPC3-DOTA-Gd, serum was collected at 24 h post injection, and the levels of pro-inflammatory cytokines (TNF-α, IL-6, IFN-γ, IL-1β) and anti-inflammatory cytokines (IL-10, TGF-β) were measured by ELISA (Figure 6). The 24 h time point was chosen because it corresponds to the early circulation phase of antibody probes, when serum concentrations are still sufficiently high to induce innate cytokine responses, but beyond the immediate infusion period. For murine antibodies such as 1G12, the reported half-life in mice is ~1–2 days and in humans ~6–12 h due to immunogenic clearance, whereas fully human IgG1 antibodies such as ET58 typically show half-lives of ~2–3 days in mice and ~2–3 weeks in humans [14,16,17]. Thus, 24 h provides a sensitive and physiologically relevant window to detect systemic immune activation while both probes remain bioavailable.

Compared to PBS and DOTA-Gd controls, both 1G12-DOTA-Gd or ET58-GPC3-DOTA-Gd caused elevated levels of IL-6, IFN-γ, and TNF-α, suggesting immune activation in response to both probes. In addition, increases in IL-10 and TGF-β were observed, indicating engagement of compensatory anti-inflammatory pathways. Notably, 1G12-DOTA-Gd caused a marked increase in IL-6 compared to ET58-DOTA-Gd. Since both probes share a murine IgG1 Fc, this observation likely reflects Fab-dependent factors rather than FcγR differences. For example, 1G12 and ET58 recognize distinct epitopes, and stronger or more accessible epitope engagement by 1G12 may increase cell-surface crosslinking, amplifying downstream pro-inflammatory signaling (e.g., IL-6) in tumor-associated or myeloid cells. Additionally, the human F(ab’)_2_ of ET58 may exhibit reduced cross-reactivity with murine co-receptors or accessory proteins compared with the fully murine 1G12 Fab, yielding a lower agonistic potential and attenuated acute cytokine release at the 24 h innate window.

Overall, ET58-DOTA-Gd elicited a weaker magnitude of cytokine elevation, whereas IG12-DOTA-Gd elicited marked elevations of cytokine levels. These findings suggest that while both probes induce a biologically detectable immune response, the chimeric ET58-based probe is associated with far lower immunogenicity, highlighting its favorable biosafety profile for clinical translation.

## 4. Discussion

Anti-GPC3 antibodies have previously been explored for imaging using radionuclides (e.g., ^89^Zr-labeled antibodies for PET) or nanoparticles (iron oxide, optical probes) [8,15,18,19]. However, no prior study has reported GPC3 antibodies conjugated with DOTA–gadolinium for MRI. Our work therefore represents the first demonstration of a GPC3-targeted DOTA-Gd probe for MRI in HCC cell lines and animal models. We successfully synthesized and validated novel GPC3-targeted MRI probes that comprise a mouse or chimeric anti-GPC3 antibody conjugated to the gadolinium-based chelator DOTA-Gd. Both probes retained specific binding to GPC3-positive HCC cells and exhibited enhanced T_1_ relaxivity in vitro and selectively identified and accumulated in GPC3-expressing HCC cells in vivo. Importantly, ET58-DOTA-Gd, comprising the chimeric anti-GPC3 antibody, displayed minimal immune activation, reinforcing its safety and suitability for clinical translation. These results support the use of antibody-based MRI not only for tumor identification but as a non-invasive tool to stratify HCC patients for GPC3-targeted therapies.

In both orthotopic and subcutaneous xenograft models, ET58-DOTA-Gd produced tumor-specific signal enhancement, in terms of intensity as well as selective prolonged retention of signals in GPC3-positive xenografts but not in GPC3-negative xenografts. The prolonged signal retention in GPC3-positive tumors, even after initial contrast from perfusion dissipates, suggests a mechanism of receptor-mediated binding and internalization, which therefore allows diagnostic differentiation of HCC tumors with varying levels of GPC3 expression [8]. In clinical cases where multiple hepatic nodules are present, the persistence of contrast enhancement over time may serve as a functional indicator of molecular target expression, providing crucial differentiation between HCC and benign lesions, or between aggressive and indolent subtypes [20,21,22].

As GPC3-targeted therapies such as CAR-T cells [23,24,25] and bispecific antibodies progress through clinical trials (NCT02748837), the need for non-invasive, reliable tools to identify suitable patients becomes increasingly urgent. Our data suggest that GPC3-targeted MRI could serve as a companion diagnostic tool, facilitating selection of GPC3-positive HCC subgroups eligible for GPC3-targeted immunotherapies or cell therapies (e.g., Phase I trial, NCT02748837), as well as longitudinal monitoring of treatment response and tumor recurrence. The minimal immune activation observed with ET58-DOTA-Gd further strengthens its potential for clinical use, especially in repeated imaging settings where cumulative immunogenicity may otherwise be a concern. Differences in Fab recognition and epitope engagement are known to influence cytokine responses [26,27,28]. Indeed, although ET58 and 1G12 both contain a murine IgG1 Fc (minimizing Fc-based disparities in mice), their differences in the F(ab’)_2_ regions might affect epitope recognition and avidity, leading to differences in receptor clustering and downstream signaling. The stronger IL-6 signal elicited by 1G12-DOTA-Gd may be attributed to Fab/epitope-driven innate activation, yielding higher IL-6 levels in response to 1G12. The human F(ab′)_2_ of ET58 may also interact less efficiently with murine accessory pathways, resulting in weaker acute cytokine induction. This feature, together with its high specificity for GPC3, highlights ET58-DOTA-Gd as a promising probe with a favorable biosafety profile for translation.

Various formats of targeting ligands have been investigated for molecular MRI, including peptides, single-chain variable fragments (scFvs), and full-length antibodies. Peptide- or scFv-based probes offer rapid blood clearance and favorable tumor-to-background ratios at early time points, making them attractive for fast imaging workflows [29,30]. In contrast, full-length antibodies such as ET58 offer high binding avidity through bivalent interactions and slower dissociation kinetics, resulting in greater target specificity and prolonged tumor retention. These attributes of full-length antibodies not only support delayed but more specific imaging, but also increased diagnostic confidence, which is especially important in tumors with heterogeneous or low-level target expression. As such, probes based on full-length antibodies are better suited for applications such as imaging-guided molecular stratification of patients, and monitoring of personalized therapy, where diagnostic accuracy is prioritized over rapid clearance.

While our statistical analyses demonstrated significant differences between groups (e.g., enhanced CNR with ET58-DOTA-Gd vs. DOTA-Gd at 40-, 90-, and 120 min post injection, *p* < 0.05; Figure 4d), we acknowledge that the sample size in this exploratory study was modest (*n* = 3 per group). The analyses were conducted using standard approaches (two-way ANOVA with Tukey’s post hoc test or Student’s *t*-test as appropriate), and the consistent effect sizes across replicates support the robustness of the findings. Nonetheless, future studies with larger sample numbers will be necessary to confirm the statistical power and generalizability of our results.

Our findings are consistent with and extend upon prior work in antibody-based imaging of HCC. Previous studies using ^89^Zr-labeled antibodies for PET imaging have demonstrated targeting specificity but are limited by radiation exposure and cost [8,9]. Other MRI probes targeting integrins [29,30,31] or VEGF [32,33,34] have been explored, but their specificity for HCC is lower than that of GPC3 [35,36]. By using a chimeric anti-GPC3 antibody, our probe offers improved translational potential compared to fully murine antibodies [37]. Compared to other antibody-based imaging modalities, our probe offers the advantage of high spatial resolution of MRI and absence of ionizing radiation, making it especially suitable for repeated use in high-risk cancer screening, and monitoring of tumor response and recurrence. While PET imaging with radiolabeled antibodies has shown promise in GPC3-positive HCC [38,39,40], it is costly and carries the risk of radiation exposure; additionally, PET imaging probes face higher regulatory hurdles [41]. In contrast, a GPC3-specific MRI probe comprising the currently used Gd contrast agent may be more readily integrated into clinical workflows and may even be used in combination with functional MRI techniques to provide a comprehensive assessment of tumor biology [42,43].

Moving forward, combining GPC3-targeted MRI with therapy response biomarkers or imaging-guided biopsy could further enhance clinical decision-making by improving diagnostic accuracy and ensuring molecular characterization is obtained from the most relevant areas of the tumor. However, we acknowledge several limitations of this work. While our imaging studies demonstrated tumor targeting and favorable contrast kinetics, they were conducted in immunodeficient mouse models that do not fully recapitulate the complexity of human immune responses or tumor microenvironments. To further reduce the potential for immunogenicity in humans, we plan to use just the human anti-GPC3 F(ab’)_2_ portion (alone or engrafted onto human IgG1 Fc) for subsequent evaluation in our orthotopic HCC PDX models, and for pharmacokinetics and toxicity studies of anti-GPC3 F(ab’)_2_ and its resulting conjugate with DOTA-Gd, especially under repeated dosing conditions. Notably, F(ab’)_2_ fragments have already been successfully applied in preclinical GPC3-targeted PET imaging studies, supporting their feasibility for translational development [44]. Lastly, while MRI provides excellent spatial resolution, its sensitivity remains lower than that of nuclear imaging, which may limit detection of micrometastases or lesions with low GPC3 expression. Nonetheless, our findings lay an important foundation for further development of antibody-based molecular MRI in HCC and support the advancement of GPC3-targeted imaging as a companion diagnostic tool in the era of personalized oncology.

## 5. Conclusions

In summary, we demonstrated that ET58, a highly specific chimeric anti-GPC3 antibody, can be successfully conjugated with standard MRI contrast agent for the safe and non-invasive detection of GPC3-positive HCC tumors. ET58 has the potential to be optimized for further clinical translation as a companion diagnostic and treatment monitoring tool for personalized therapies targeting GPC3-positive HCC patients.

Overall, our study highlights the potential of using targeted MRI probes to complement existing imaging modalities, enabling more accurate assessment of tumor biology and treatment response. Such approaches may ultimately bridge molecular imaging and personalized therapy, paving the way for broader clinical translation in HCC and other solid tumors.

## Figures and Tables

**Figure 1 cancers-17-03357-f001:**
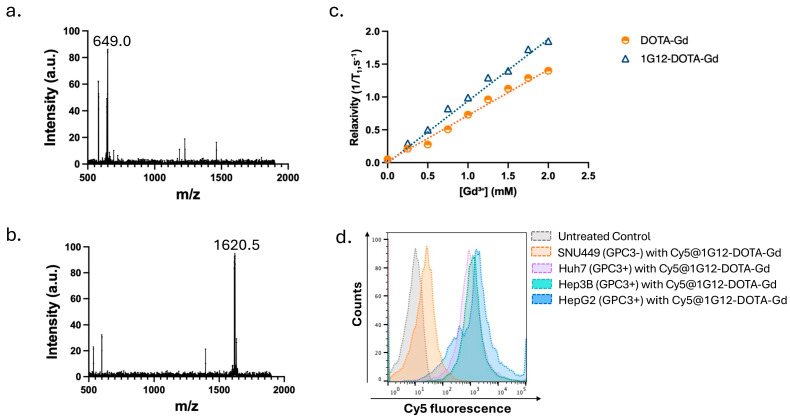
Characterization of conjugation, relaxivity, and target binding of 1G12-DOTA-Gd. (**a**) Mass spectrometry analysis (*m*/*z* measurement) of DOTA-Gd, and (**b**) 1G12-DOTA-Gd. (**c**) Relaxivity (r_1_) measurement of 1G12-DOTA-Gd at 3T MRI shows higher relaxivity compared to non-targeted DOTA-Gd across [Gd^3+^] concentrations, indicating enhanced T_1_ contrast potential. (**d**) Flow cytometry with Cy5-labeled 1G12-DOTA-Gd shows selective binding to HepG2, Hep3B, Huh7 cell lines (GPC3+ve) and reduced binding to SNU449 cells (GPC3−ve).

**Figure 2 cancers-17-03357-f002:**
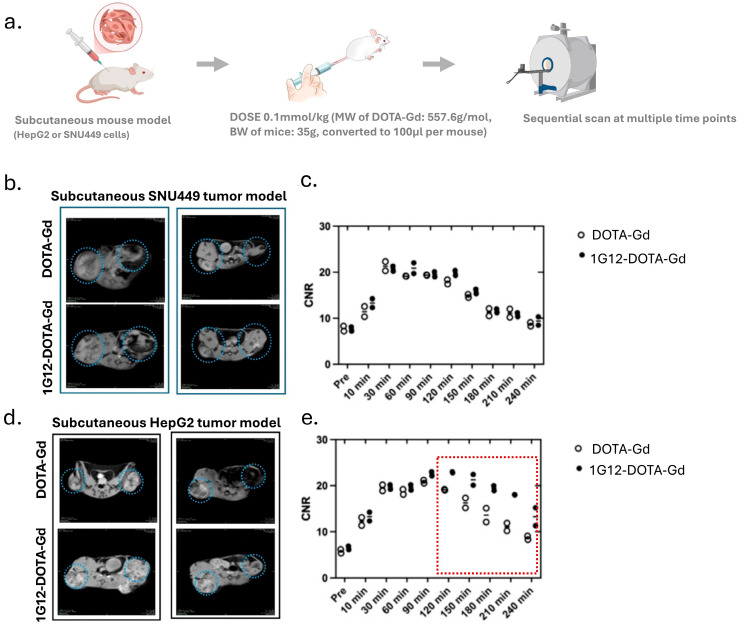
MRI images and CNR analysis in subcutaneous tumor models after injection with 1G12-DOTA-Gd. (**a**) Schematic illustration of imaging workflow in subcutaneous tumor models. (**b**) Representative T_1_-weighted MRI images. The blue dashed circles indicate the tumor regions analyzed for contrast-to-noise ratio (CNR) quantification. (**c**) corresponding CNR values over time, post injection of SNU449 (GPC3−ve) tumor models. (**d**) Representative T_1_-weighted MRI images. The blue dashed circles indicate the tumor regions analyzed for contrast-to-noise ratio (CNR) quantification. (**e**) corresponding CNR values over time, post injection of HepG2 (GPC3+ve) tumor models. The dashed box highlights the time window showing the most pronounced CNR difference between 1G12-DOTA-Gd and DOTA-Gd.

**Figure 3 cancers-17-03357-f003:**
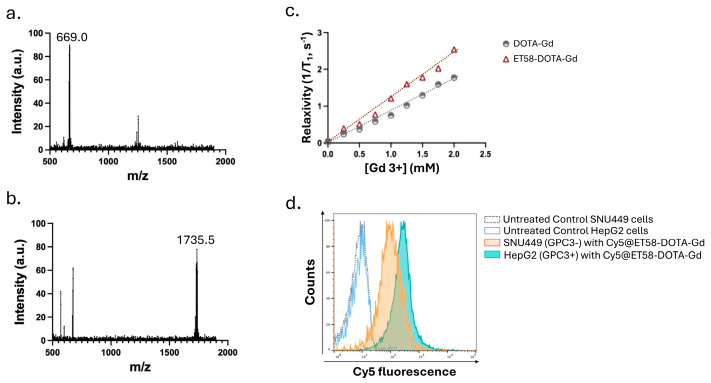
Characterization of conjugation, relaxivity, and target binding of ET58-DOTA-Gd. Mass spectrometry analysis (*m*/*z* measurement) of (**a**) DOTA-Gd, and (**b**) ET58-DOTA-Gd confirm successful conjugation of ET58 with DOTA-Gd, with a *m*/*z* shift from 669.0 to 1735.5. (**c**) Relaxivity (r_1_) measurement of DOTA-Gd and ET58-DOTA-Gd at 3T across 0–2.0 mM [Gd^3+^]; ET58-DOTA-Gd shows higher relaxivity. (**d**) FACS analysis of Cy5-labeled ET58-DOTA-Gd shows specific binding to HepG2 cells (GPC3+ve), with weaker signal in SNU449 (GPC3−ve) and untreated cells used as controls.

**Figure 4 cancers-17-03357-f004:**
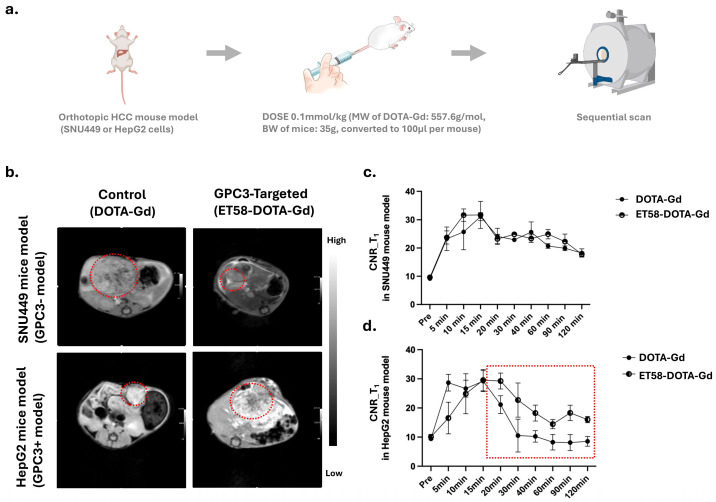
MRI images and CNR analysis in orthotopic tumor models after injection with ET58-DOTA-Gd. (**a**) Schematic of imaging workflow in orthotopic HCC mouse models. Representative T_1_-weighted MRI images of orthotopic liver tumors in (**b**) GPC3-negative (SNU449, upper panel) and GPC3-positive (HepG2, lower panel) following injection of DOTA-Gd or ET58-GPC3-DOTA-Gd. CNR values over time in (**c**) SNU449 or (**d**) HepG2 orthotopic xenografts, respectively, post injection with DOTA-Gd or ET58-GPC3-DOTA-Gd. The red dashed circles indicate the tumor regions analyzed for CNR quantification. Bars represent mean ± SEM (*n* = 3). Significant differences between ET58-DOTA-Gd and DOTA-Gd groups in HepG2 mouse models were observed at 40, 90, and 120 min (*p* < 0.05, two-way ANOVA with Tukey’s post hoc test. The dashed box highlights the time window showing the most pronounced CNR difference between ET58-DOTA-Gd and DOTA-Gd.

**Figure 5 cancers-17-03357-f005:**
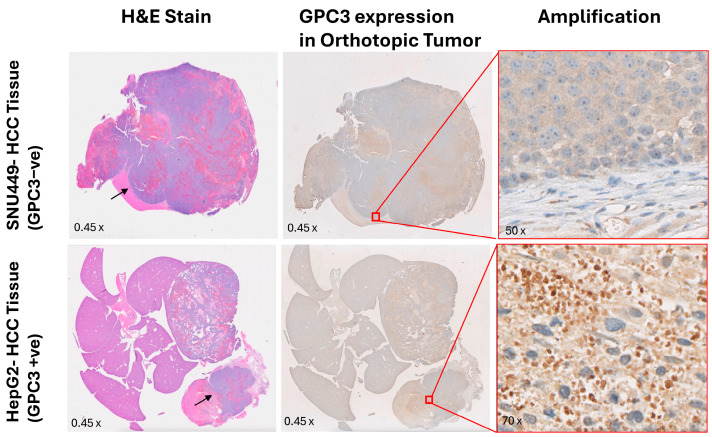
H&E staining and IHC staining for GPC3 expression in SNU449 (GPC3-negative, **upper** panel) and HepG2 (GPC3-positive, **lower** panel) xenografts. **Left** Panel (H&E Staining) shows tumor morphology of SNU449 (**top** row) and HepG2 (**bottom** row) xenografts encompassing xenograft and surrounding liver tissues. Middle Panel and its amplification (**right** panel) show IHC staining for GPC3 expression: SNU449 xenograft (**top** row) shows negligible staining confirming low or absent GPC3 expression, whereas HepG2 xenograft (**bottom** row) shows strong cytoplasmic and membranous GPC3 staining, confirming high GPC3 expression in almost all tumor cells in the observed area. Black arrows indicate the border between xenografts and adjacent liver tissues.

**Figure 6 cancers-17-03357-f006:**
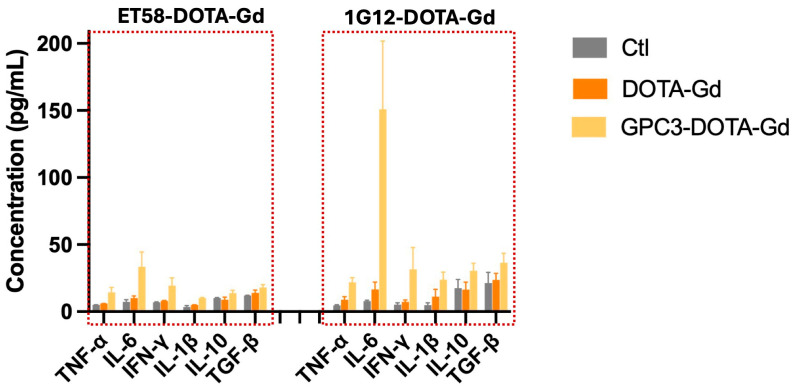
Serum cytokine levels following administration of ET58-DOTA-Gd or 1G12-DOTA-Gd. Serum levels of pro-inflammatory (TNF-α, IL-6, IFN-γ, IL-1β) and anti-inflammatory (IL-10, and TGF-β) cytokines measured by ELISA at 24 h post injection of ET58-DOTA-Gd, 1G12-DOTA-Gd, DOTA-Gd, or PBS (control). Bars represent mean ± SD (*n* = 3), Bars = mean ± SD (*n* = 3). Significance is defined as *p* < 0.05, using two-way ANOVA with Tukey’s post hoc test. Red dashed boxes indicate the experimental groups (ET58-DOTA-Gd and 1G12-DOTA-Gd) that were compared for cytokine profiling.

## Data Availability

The data that support the findings of this study are available from the corresponding author upon request.

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
