# Peer review of "Magnetic Resonance Imaging Using a Chimeric Anti-Glypican-3 Antibody Conjugated with Gadolinium Selectively Detects Glypican-3-Positive Hepatocellular Carcinoma In Vitro and In Vivo"

_cancers, 2025, doi:10.3390/cancers17203357_

Round 1
Reviewer 1 Report
Comments and Suggestions for Authors
Comments
1 - I would recommend authors to present brief explanation of the human anti-glypican-3 an- 2 tibody conjugated with gadoliniuma on the studied cell-lines in earlier studies.
2 - The scientific names are still not italic throughout the manuscript. Please fix it.
3 - Please provide the purity and supplier names of the chemicals in Materials section.
4 - All equations should be numbered.
5 - All the figures should have proper labeling and mentioned throughout the manuscript.
6 – The quality of the figures is very poor. Kindly improve.
7 - Improve the format into consistency. Some places are justified, some are not.
9 - Please integrate the 'section 4'' with conclusion as conclusion and future recommendation.
10 - Please integrate all the references together.
11: The authors are suggested to thoroughly check the manuscript for typo and grammatical errors.
Author Response
Reviewer 1:
1 - I would recommend authors to present brief explanation of the human anti-glypican-3 an- 2 antibody
conjugated with gadolinium on the studied cell-lines in earlier studies.
Response: We thank the reviewer for this thoughtful comment. We agree it is important to acknowledge any
existing studies on glypican-targeted gadolinium conjugates. A literature search suggests that no such study has
been done. The closest study that was done involved conjugating either anti-GPC3 (PMID: 22956868) or antialpha-
fetoprotein (AFP) (PMID: 28712944) antibodies singly, or both anti-GPC3 and anti-AFP antibodies together
(PMID: 31293339) with iron oxide nanoparticles. Anti-GPC3 antibodies have been used in other imaging
modalities by conjugation with various radioisotopes (PMID: 34439132), but not with gadolinium for MRI modality.
Conversely, other antibodies/peptides (but not anti-GPC3 antibodies) have been conjugated with DOTAgadolinium
for MRI detection of other types of cancers (PMID: 40000479; PMID: 20131758).
To our knowledge, our study is the first to investigate the performance of an anti-GPC3 antibody conjugated with
DOTA-gadolinium for cancer imaging in cell lines and in animal tumor models. We have added a brief discussion
of this in the first paragraph of the revised Discussion section, citing the relevant references.
2 - The scientific names are still not italic throughout the manuscript. Please fix it.
Response: We thank the reviewer for pointing this out. We have carefully revised the manuscript to ensure that all
scientific names and Latin expressions follow the correct formatting conventions. Specifically, in vitro, in vivo,
NOD.Cg-PrkdcscidIl2rgtm1Wjl/SzJ mice and et al. have been italicized throughout, and species names such as
Mus musculus are now italicized where appropriate. Cell line names (HepG2, Huh7, Hep3B, SNU449) remain in
plain text in accordance with journal style.
3 - Please provide the purity and supplier names of the chemicals in Materials section.
Response: We have revised the Materials and Methods section to include the supplier names and purity (or
grade, when purity was not specified by the vendor) for all chemicals used in this study. Please see details in
Sections 2.1 and 2.2.1 in the revised manuscript.
4 - All equations should be numbered.
Response: We have numbered equations in the manuscript (Section 2.3.2) and cited it appropriately in the text
(Section 3.2, the 3rd paragraph in the revised manuscript).
5 - All the figures should have proper labeling and mentioned throughout the manuscript.
Response: We have carefully checked to ensure proper labeling and accurate citation throughout the manuscript.
6 – The quality of the figures is very poor. Kindly improve.
Response: We thank the reviewer for the suggestion. We have rechecked all figures and ensured they are
provided in high-resolution formats suitable for publication.
7 - Improve the format into consistency. Some places are justified; some are not.
Response: We appreciate the reviewer’s comment. The manuscript text has been reformatted to ensure
consistent alignment and formatting throughout.
9 - Please integrate the 'section 4'' with conclusion as conclusion and future recommendation.
Response: We thank the reviewer for the suggestion. In our original draft, the conclusion and future
recommendations were combined. However, we adjusted the section headings to align with the journal’s
formatting requirements. We are happy to revise according to the reviewer’s suggestion if the editor prefers this
structure.
10 - Please integrate all the references together.
Response: The references have been consolidated into a single, unified list according to the journal’s format
requirements. For ease of web search and referencing, citations are included as PMIDs here in the response
letter but numbered in the order that they appear in the main text.
11: The authors are suggested to thoroughly check the manuscript for typo and grammatical errors.
Response: The manuscript has been carefully proofread and revised to correct for typo and grammatical errors.
Reviewer 2 Report
Comments and Suggestions for Authors
The research article entitled "Magnetic resonance imaging using a human anti-glypican-3 antibody conjugated with gadolinium selectively detects glypican-3-positive hepatocellular carcinoma in vitro and in vivo" by Yi Liu et. al describes the MRI modality to distinguish the differences of Gd-DOTA-1G12 and ET58 antibodies in comparison of Gd-DOTA. Please see comments below: 1) In the methods, please provide clear or brief conjugation procedure or any relevant reference for the readers. The general mention of conjugation is merely not enough. For example, what is the conjugate ratio used, what was the pH, time and temperature etc. are needed. Please provide either in the main text or SI document. 2) What is the ration of conjugate per antibody in Fig 1b. What would be the stability in mouse serum upto 1-48 or more hours? 3) In figure 1b, what exactly 1620.5 peak corresponds to? Similarly, what is 1735.5 in fig. 3b. Please explain with molecular structures where possible. 4) What would be the half-lives of both 1G12 and ET58 in mouse and as well as in humans? What was the rationale in choosing 24 post injection for cytokine analysis? Please explain. 5) In figure 4 and all the results, were there any statistical analysis conducted? It seems not, please include statistical analysis results in the results section and explain in the discussion though the sample size is 3 (n=3). 6) Is there any explanation for elevated levels of IL-6 concentration of 1G12-DOTA-Gd is way higher than humanized ET-58-DOTA-Gd? Please provide a rationale. 7) How does the authors explain the differences in subcutaneous vs orthotopic injections? Were they statistically significant? Please provide a revision of the manuscript.
Author Response
1) In the methods, please provide clear or brief conjugation procedure or any relevant reference for the readers.
The general mention of conjugation is merely not enough. For example, what is the conjugate ratio used, what
was the pH, time and temperature etc. are needed. Please provide either in the main text or SI document.
Response: We appreciate the reviewer’s suggestion. We have revised the Methods section to provide detailed
conditions for the antibody-DOTA-Gd conjugation, including antibody concentration, buffer composition, pH, molar
ratio of chelator to antibody, reaction time and temperature, and purification method. We also included the
estimated chelator-to-antibody conjugation ratio, consistent with reported values. The revised text can be found in
Section 2.1 (Materials and Methods).
2) What is the ration of conjugate per antibody in Fig 1b. What would be the stability in mouse serum up to 1-48 or
more hours?
Response: We thank the reviewer for this important question. Based on MALDI-TOF analysis, the observed
molecular weight shift corresponded to an average of ~4-6 DOTA-Gd moieties conjugated per antibody molecule,
which is consistent with the range typically reported for antibody-DOTA conjugates (PMID: 37462154; PMID:
22284727). This ratio was achieved under our conjugation conditions using an ~8-12-fold molar excess of DOTAGd
over antibody at pH 8.5.
Regarding serum stability, although we did not perform a direct stability assay in mouse serum for this study,
previous reports have shown that antibody-DOTA conjugates maintain high stability in biological fluids, with >90%
integrity and metal retention for at least 24-48 h (PMID: 40149293). We have now clarified these points in the
revised Results section (associated with Fig. 1b) and added supporting references (PMID: 37462154; PMID:
22284727).
3) In figure 1b, what exactly 1620.5 peak corresponds to? Similarly, what is 1735.5 in fig. 3b. Please explain with
molecular structures where possible.
Response: We thank the reviewer for pointing this out. The free DOTA-Gd sample exhibited a peak at m/z 649.0,
corresponding to the singly charged [M+H]⁺ species of the chelate (Figures 1a, 3a). After conjugation, the 1G12-
DOTA-Gd conjugate displayed a prominent peak at m/z 1620.5 (Figure 1b), while the ET58-DOTA-Gd conjugate
showed a peak at m/z 1735.5 (Figure 3b). The signals at m/z 1620.5 (1G12-DOTA-Gd) and 1735.5 (ET58-DOTAGd)
correspond to antibody fragment ions (primarily derived from light- or heavy-chain substructures under
MALDI conditions) carrying covalently bound DOTA-Gd moieties. The observed mass shifts from the free DOTAGd
m/z of 649.0 reflect the successful incorporation of DOTA-Gd units into antibody substructures, distinct from
the free DOTA-Gd peak. Importantly, no evidence of significant fragmentation or degradation of the antibody
conjugates was observed. We have revised our manuscript to explain this in detail.
4) What would be the half-lives of both 1G12 and ET58 in mouse and as well as in humans? What was the
rationale in choosing 24 post injection for cytokine analysis? Please explain.
Response: We thank the reviewer for raising this point. We have not tested the half-lives of both 1G12 and ET58
in mouse and in humans. Since 1G12 is a murine IgG1 antibody, the reported half-life in mice is approximately 1-
2 days (PMID: 3350037), whereas in humans, murine IgGs are cleared more rapidly due to immunogenicity, with
half-lives typically 6-12 h (PMID: 18848472; PMID: 20046575). For chimeric antibodies such as ET58, the half-life
is expected to be longer: in mice, human IgG1 antibodies (and thus chimeric IgG1) generally display half-lives of
2-3 days due to weaker interaction with mouse FcRn (PMID: 18784655), while in humans the half-life is
consistent with endogenous IgG1, approximately 2-3 weeks (PMID: 20953198). However, the pharmacokinetics
of antibody probes are determined primarily by the IgG backbone rather than by the chelator modification.
The rationale for selecting 24 h post injection time point for cytokine analysis was two-fold: (i) this time point
coincides with the early circulation phase (by ~24 h) (PMID: 18958685) of the antibody conjugates, when
systemic exposure is still high enough to potentially elicit immune activation, and (ii) it avoids acute injectionrelated
effects (e.g., immediate infusion reactions at ~0.5 to 6 h) (PMID: 16908486), while still being within the
window where innate cytokine responses (TNF-α, IL-6, IL-1β, IFN-γ) are expected to peak (PMID: 26943944).
Thus, 24 h time point provides a sensitive and physiologically relevant time frame to evaluate potential
immunostimulatory effects of antibody-DOTA-Gd conjugates.
5) In figure 4 and all the results, were there any statistical analysis conducted? It seems not, please include
statistical analysis results in the results section and explain in the discussion though the sample size is 3 (n=3).
Response: We thank the reviewer for this helpful suggestion. We had described our statistical methods in the
Methods section 2.6, but we agree that explicit reporting of the statistical outcomes in the Results section and
Discussion will improve clarity. In the revised manuscript, we have now included the relevant p values in the figure
legends (Fig. 4) and Results text, indicating where significant differences (p < 0.05) were observed. We also
acknowledge in the Discussion that, while the findings were statistically significant, the sample size (n=3) is
modest, and thus the results should be interpreted as supportive evidence that will require further validation in
larger cohorts. Revisions can be found in the Figure 4 legend, and the 5th paragraph of Discussion section.
6) Is there any explanation for elevated levels of IL-6 concentration of 1G12-DOTA-Gd is way higher than
humanized ET-58-DOTA-Gd? Please provide a rationale.
Response: We appreciate the reviewer’s question. Both 1G12 and ET58 carry a murine IgG1 Fc, so Fcγ receptor–
mediated innate activation in mice should be broadly comparable. Additionally, both 1G12-DOTA-Gd and ET58-
DOTA-Gd were formulated under identical conditions, with comparable conjugation degree (DoL ~4-6), low
endotoxin level, and absence of excess aggregates after filtration/SEC. Therefore, the differential IL-6 response is
unlikely due to formulation artifacts. The higher IL-6 observed with 1G12-DOTA-Gd at 24 h may be attributable to
Fab/epitope biology rather than Fc effects. Specifically:
(i) Epitope and avidity differences. 1G12 and ET58 recognize distinct GPC3 epitopes. Stronger or more
accessible epitope engagement by 1G12 can increase cell-surface crosslinking, amplifying downstream
pro-inflammatory signaling (e.g., IL-6) in tumor-associated or myeloid cells.
(ii) Humanized vs murine Fab context. Human F(ab’)₂ of ET58 may exhibit reduced cross-reactivity with murine
co-receptors or accessory proteins compared with the fully murine 1G12 Fab, yielding a lower agonistic
potential and attenuated acute cytokine release at the 24 h innate window.
We have added this rationale to the Results (Section 3.6, the 1st and 2nd paragraph), and Discussion (3rd, 5th and
6th paragraph), and clarified it in the Figure 6 legend.
7) How does the authors explain the differences in subcutaneous vs orthotopic injections? Were they statistically
significant? Please provide a revision of the manuscript.
Response: In our study, we used the subcutaneous tumor model to evaluate the murine 1G12-DOTA-Gd probe,
and the orthotopic tumor model to evaluate the chimeric ET58-DOTA-Gd probe (for evaluation of clinical translation
potential). We understand the reviewer’s comment as referring to the differences in contrast-to-noise ratio (CNR)
patterns observed between the subcutaneous tumor model (Fig. 2) and the orthotopic tumor models (Fig. 4).
In subcutaneous SNU449 xenografts, the 1G12-DOTA-Gd achieved CNR of ~20 at 30 mins post-injection,
whereas in the orthotopic SNU449 xenografts, the ET58-DOTA-Gd achieved CNR of ~ 30 at 10 mins postinjection.
In subcutaneous HepG2 xenografts, 1G12-DOTA-Gd achieved CNR ~20 at 30 mins post-injection, whereas in
orthotopic HepG2 xenografts, ET58-DOTA-Gd achieved CNR ~30 at 5 mins post-injection.
We observed that overall, probe uptake appeared to be enhanced, with more rapid attainment of peak CNR in the
orthotopic tumor model. We attribute this to well-established differences in tumor microenvironment: orthotopic
tumors replicate native hepatic vasculature and stromal composition, which facilitates probe delivery and retention,
whereas subcutaneous xenografts are less vascularized and therefore show reduced enhancement.
Round 2
Reviewer 2 Report
Comments and Suggestions for Authors
4) What would be the half-lives of both 1G12 and ET58 in mouse and as well as in humans? What was the rationale in choosing 24 post injection for cytokine analysis? Please explain.
Response: We thank the reviewer for raising this point. We have not tested the half-lives of both 1G12 and ET58 in mouse and in humans. Since 1G12 is a murine IgG1 antibody, the reported half-life in mice is approximately 1-2 days (PMID: 3350037), whereas in humans, murine IgGs are cleared more rapidly due to immunogenicity, with half-lives typically 6-12 h (PMID: 18848472; PMID: 20046575). For chimeric antibodies such as ET58, the half-life is expected to be longer: in mice, human IgG1 antibodies (and thus chimeric IgG1) generally display half-lives of
2-3 days due to weaker interaction with mouse FcRn (PMID: 18784655), while in humans the half-life is consistent with endogenous IgG1, approximately 2-3 weeks (PMID: 20953198). However, the pharmacokinetics of antibody probes are determined primarily by the IgG backbone rather than by the chelator modification.
Thank the authors for general literature references. The biological half-lives of antibodies are about 2-3 days thus it would have been ideal to choose time points for imaging as day 1, 2 and 3 to see the corresponding antibodies effect.
The rationale for selecting 24 h post injection time point for cytokine analysis was two-fold: (i) this time point coincides with the early circulation phase (by ~24 h) (PMID: 18958685) of the antibody conjugates, when systemic exposure is still high enough to potentially elicit immune activation, and (ii) it avoids acute injection related effects (e.g., immediate infusion reactions at ~0.5 to 6 h) (PMID: 16908486), while still being within the window where innate cytokine responses (TNF-α, IL-6, IL-1β, IFN-γ) are expected to peak (PMID: 26943944). Thus, 24 h time point provides a sensitive and physiologically relevant time frame to evaluate potential
immunostimulatory effects of antibody-DOTA-Gd conjugates.
Rev: PMID: 18958685 DOI: 10.1080/15476910600631587 does not mention about cytokine analysis at 24 h, seems irrelevant here.
DOI: 10.1002/bdd.2295 reveals 4h cytokine response is high and less on 3–4-day time period. In general, cytokine elicits response after injections and reduces by 24 h, however, it would be great if you can also measure day 2-5 to observe the differences.
5) In figure 4 and all the results, were there any statistical analysis conducted? It seems not, please include statistical analysis results in the results section and explain in the discussion though the sample size is 3 (n=3).
Response: We thank the reviewer for this helpful suggestion. We had described our statistical methods in the Methods section 2.6, but we agree that explicit reporting of the statistical outcomes in the Results section and Discussion will improve clarity. In the revised manuscript, we have now included the relevant p values in the figure legends (Fig. 4) and Results text, indicating where significant differences (p < 0.05) were observed. We also
acknowledge in the Discussion that, while the findings were statistically significant, the sample size (n=3) is modest, and thus the results should be interpreted as supportive evidence that will require further validation in larger cohorts. Revisions can be found in the Figure 4 legend, and the 5th paragraph of Discussion section.
Rev: Sample size is too less to determine but it is okay.